# Socio-demographic and genetic risk factors for drug adherence and persistence across 5 common medication classes

Mattia Cordioli [1,13], Andrea Corbetta [1,2,3,13], Hanna Maria Kariis[4,13], Sakari Jukarainen [1], Pekka Vartiainen [1], Tuomo Kiiskinen [1], Matteo Ferro[1], FinnGen*, Estonian Biobank Research Team*, Markus Perola[5], Mikko Niemi [6,7,8], Samuli Ripatti [1,9,10], Kelli Lehto[4], Lili Milani[4] & Andrea Ganna [1,11,12] ✉

Low drug adherence is a major obstacle to the benefits of pharmacotherapies and it is therefore important to identify factors associated with discontinuing or being poorly adherent to a prescribed treatment regimen. Using high-quality nationwide health registry data and genome-wide genotyping, we evaluate the impact of socio-demographic and genetic risk factors on adherence and persistence for 5 common medication classes that require long-term, regular therapy ($N = 1,814,591$ individuals from Finnish nationwide registries, 217,005 with genetic data from Finland and Estonia). Need for social assistance and immigration status show a notable negative effect on persistence and adherence across the examined medications (odd ratios between 0.48 and 0.82 for persistence and between 1.1% to 4.3% decrease in adherence) while demographic and health factors show comparably modest or inconsistent effects. A genome-wide scan does not identify genetic variants associated with the two phenotypes, while some pharmacogenes (i.e. *CYP2C9* and *SLCO1B1*) are modestly associated with persistence, but not with adherence. We observe significant genetic correlations between medication adherence and participation in research studies. Overall, our findings suggest that socio-economically disadvantaged groups would benefit from targeted interventions to improve the dispensing and uptake of pharmacological treatments.

Drug adherence is defined as the extent to which patients follow the prescribed therapeutic regimen when taking medications[1]. High adherence is considered to be crucial for the effectiveness of pharmacological treatments[2], and improving adherence has been considered one of the leading challenges for healthcare professionals[3]. A better understanding of the risk factors driving adherence is fundamental to helping clinicians improve their clinical practice and identify patients who are at risk of poor adherence at an earlier stage.

[1]Institute for Molecular Medicine Finland, University of Helsinki, Helsinki, Finland. [2]CHDS - Health Data Science Center, Human Technopole, Milan, Italy. [3]MOX - Laboratory for Modeling and Scientific Computing, Department of Mathematics, Politecnico di Milano, Milan, Italy. [4]Estonian Genome Centre, Institute of Genomics, University of Tartu, Tartu, Estonia. [5]The Finnish Institute for Health and Welfare, Helsinki, Finland. [6]Department of Clinical Pharmacology, University of Helsinki, Helsinki, Finland. [7]Individualized Drug Therapy Research Program, University of Helsinki, Helsinki, Finland. [8]Department of Clinical Pharmacology, HUS Diagnostic Center, Helsinki University Hospital, Helsinki, Finland. [9]Broad Institute of MIT and Harvard, Cambridge, MA, USA. [10]Department of Public Health, University of Helsinki, Helsinki, Finland. [11]Analytic and Translational Genetics Unit, Massachusetts General Hospital, Boston, MA, USA. [12]Program in Medical and Population Genetics, Broad Institute of MIT and Harvard, Cambridge, MA, USA. [13]These authors contributed equally: Mattia Cordioli, Andrea Corbetta, Hanna Maria Kariis.*Lists of authors and their affiliations appear at the end of the paper. ✉e-mail: andrea.ganna@helsinki.fi

While there are multiple dimensions of adherence, some related to the healthcare system and the mode of therapy delivery[4,5], studies have shown that individuals with low adherence in placebo arms of randomized controlled trials have higher mortality[6]. This suggests that adherence to pharmacotherapy may be a surrogate marker for overall health behavior (healthy adherer effect) and that risk factors for adherence might be shared across different drugs.

Accurately estimating patients drug usage and adherence is not a trivial task. Traditionally questionnaires have been used, but with the increasing availability of electronic health records, studies have started investigating determinants of drug adherence and persistence using pharmacy administrative databases or insurance claims data[7–11]. However, most studies focus on a single medication or condition and consider adherence over a limited follow-up available due to data limitations. Few studies have investigated determinants of adherence across different medications[12,13], but have only considered a limited number of demographic and socioeconomic risk factors.

Beyond socio-demographic factors, genetics might play a role in drug adherence. Pharmacogenomic studies have identified several genetic variants relevant to drug response that can increase an individual's risk for adverse drug reactions[14], and a recent randomized trial suggested that pre-emptive testing with a 12-gene pharmacogenetic panel can significantly reduce the incidence of clinically relevant adverse drug reactions[15]. Because an adverse drug reaction decreases the likelihood of a patient continuing to take a medication, genetics might impact adherence via this mechanism. Nonetheless, the impact of clinically relevant pharmacogenetic variants on drug adherence has not yet been studied in large biobank-based studies and considering multiple medications. Moreover, identifying novel genetic variants associated with adherence might help to discover new aspects of drug response. An alternative, but not mutually exclusive hypothesis, is that genetics might capture underlying health and behavioral aspects that are associated with adherence. In this scenario, polygenic scores (PGS) for certain diseases or behavioral traits might help predict drug adherence.

In this work, we conduct a comprehensive and systematic investigation of adherence and persistence to drug treatments across multiple patient-related dimensions encompassing demographic, socioeconomic, and genetic factors. We use high-quality nationwide electronic health record data on prescription medication purchases made at pharmacies to accurately estimate two established drug-taking phenotypes[16–19] (Methods): (a) persistence, defined as purchasing medications for at least 1 year versus early discontinuation of the medication (i.e., after one purchase) and (b) adherence, measured as medication possession ratio (MPR), which is the "proportion (or percentage) of days supply obtained during a specified time period" (Fig. 1, Methods). We consider five commonly prescribed medications that are typically used for long-term, continuous, regular therapy.

We first explore the role of multiple health, demographics, and socioeconomic factors in FinRegistry[20], a nationwide cohort study. Next, we consider the role of genetics in a subset of the Finnish population from the FinnGen study[21] and in the Estonian Biobank[22], where genome-wide genotype information is available. We focus on well-known pharmacogenes as well as on genome-wide effects. We perform a genome-wide association study (GWAS) to identify potential novel genetic associations with adherence and persistence, and we

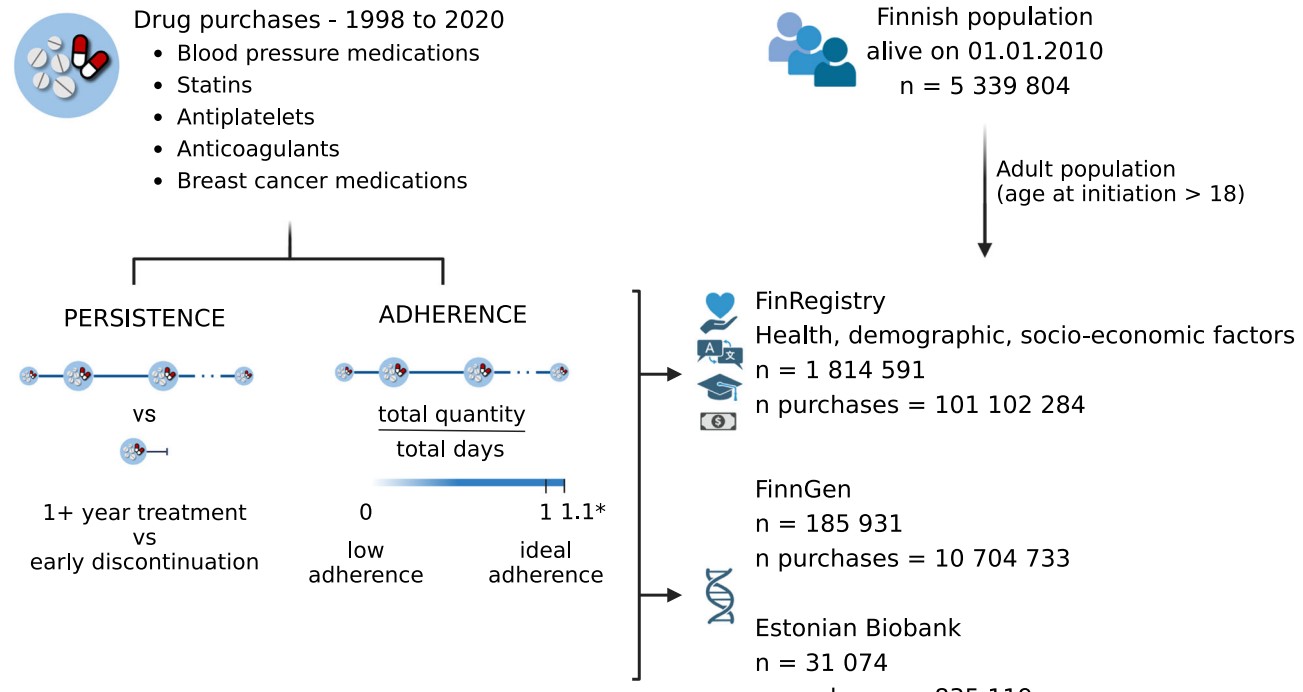

**Fig. 1 | Schematic overview of the study.** We included all individuals resident in Finland and alive on 1.1.2010 (*N* = 5,339,804) and used information from the Finnish drug purchase register to identify individuals with at least one purchase for the following medication classes: blood pressure medications, statins, antiplatelets, direct oral anticoagulants, and breast cancer medications. We considered only the adult population, starting the treatment after age 18, and reconstructed individual medication purchase trajectories as a proxy to define the medication usage phenotypes: persistence and adherence. Persistence was defined as continuing the treatment for at least 1 year versus discontinuing it after the first purchase. We defined adherence as the medication possession ratio (ratio of the quantity purchased in the observation period to the days in the observation period), for patients continuing the treatment for at least 1 year. Adherence is a continuous trait ranging from 0 (complete lack of adherence) to 1 (ideal adherence). *We excluded patients with adherence above 1.1, to exclude excessive stockpiling or overbuying behaviors, but still allowing for some variability. We then studied the association of these two medication-use phenotypes with health, demographic, and socio-economic factors from nationwide registries (*N* = 1,814,591 with complete information) and with genetic information for a subset of the population from the FinnGen study (*N* = 185,931) and from Estonian Biobank (*N* = 31,074). Created in BioRender. Cordioli, M. (2024) BioRender.com/i60h334.

**Table 1 | Descriptive statistics for the five medication classes included in the analysis**

| Medication class | ATC codes | Cohort | N individuals | Total purchases | Proportion of persistent individuals | Mean adherence (SD) | Mean treatment length (years) |
|---|---|---|---|---|---|---|---|
| Blood Pressure medications | C02*, C03*, C08*, C09* | FinRegistry | 1,316,889 | 61,213,707 | 0.947 | 0.93 (0.123) | 10.755 |
| | | FinnGen | 134,615 | 6,399,494 | 0.937 | 0.932 (0.121) | 11.172 |
| Statins | C10AA* | FinRegistry | 1,061,926 | 33,864,971 | 0.952 | 0.892 (0.123) | 9.168 |
| | | FinnGen | 116,439 | 3,828,359 | 0.956 | 0.901 (0.118) | 9.454 |
| | | Estonian Biobank | 27,339 | 792,839 | 0.866 | 0.648 (0.275) | 8.815 |
| Antiplatelets | B01AC04, B01AC30 | FinRegistry | 138,007 | 3,120,237 | — | 0.931 (0.114) | 5.528 |
| | | FinnGen | 12,618 | 285,902 | — | 0.933 (0.112) | 5.572 |
| Clopidogrel | B01AC04 | Estonian Biobank | 2712 | 33,466 | — | 0.663 (0.325) | 5.455 |
| Anticoagulants | B01AF01, B01AF02, B01AF03, B01AE07 | FinRegistry | 105,113 | 2,003,698 | 0.975 | 0.957 (0.094) | 3.127 |
| | | FinnGen | 2792 | 45,394 | 0.889 | 0.952 (0.101) | 2.984 |
| Breast Cancer medications | L02BA01, L02BG04, L02BG06, L02BG03 | FinRegistry | 55, 270 | 899,671 | 0.984 | 0.969 (0.067) | 4.556 |
| | | FinnGen | 9857 | 145,584 | 0.985 | 0.971 (0.067) | 4.128 |
| Tamoxifen | L02BA01 | Estonian Biobank | 1023 | 8814 | 0.58 | 0.829 (0.267) | 3.973 |

For each medication class, we report the ATC codes used to define the medication purchases of interest and other descriptive statistics in the general Finnish population (FinRegistry), in the subset of the Finnish population with genetic data (FinnGen) and in the Estonian Biobank. Number of individuals is the total number of individuals included in the analysis, meaning the number of individuals considered for both the persistence and adherence analysis. Total purchases is the total number of purchases per each medication and each cohort considered in both analyses. Mean adherence (and its standard deviation) and the mean treatment length are reported for the individuals with at least one year long treatment (individuals considered in the adherence analysis).

evaluate if these results can be used for risk stratification. Finally, we use genetic correlations analysis to examine the overlap between persistence and adherence and other diseases or behavioral traits, and whether PGSs capturing these traits can be used to predict the two drug-taking phenotypes of interest.

## Results

### Adherence and persistence for five medications in 1,814,591 individuals in the general population and 217,005 individuals with genetic data

We considered five classes of medications: statins, blood pressure (BP) medications, antiplatelets, breast cancer medications, and direct oral anticoagulants (DOAC) (ATC codes included are described in Table 1 and Methods). For breast cancer medications and DOACs, we included individuals with, respectively, breast cancer or atrial fibrillation, pulmonary embolism, or venous thromboembolism diagnosis before treatment initiation (Methods). For antiplatelets, we did not analyze persistence, as this class of medications is frequently prescribed for temporary treatments of 6 to 12 months. Information about the date, quantity, and type of reimbursable medications that were purchased in pharmacies were available between 1998 and 2020 and were used to calculate adherence and persistence.

Our initial study population included every individual who was a resident in Finland on 1.1.2010 (N = 5,339,804). About 2,289,455 individuals (43% of the whole population) purchased at least one of the five medications after 18 years of age, with a total of 162,231,752 purchases.

Most individuals purchased the five medications for at least 12 months (ranging from 89.7% for breast cancer medications to 64.7% for anticoagulants, except for antiplatelets (44.2%), Supplementary Data 1) and were eligible for the adherence analysis (N = 1,876,212). We further excluded 141,452 individuals who had a medication possession ratio (MPR) >1.1, to account for stockpiling of medications or over-buying behaviors (Methods), and 1270 individuals with incomplete registry information for the 8 health and socioeconomic factors considered in the main analysis (Supplementary Data 1). In the persistence analysis, we considered people discontinuing the treatment after one purchase and compared them to those continuing for at least

12 months and were eligible for the adherence analysis. Our final study population thus included 1,814,591 individuals, for a total of 101,102,284 purchases.

BP medications were the most purchased (N of individuals = 1,316,889) and had a mean treatment length of over 10 years (Table 1). Breast cancer medications were purchased by the fewest individuals (N = 55,270) and most individuals stopped after 5 years.

Persistence was above 90% for all drugs (Fig. 2). Adherence was 0.94, averaging across all medications, highest in breast cancer medications (0.97), and lowest in statins (0.89; Table 1). Adherence values were similar to those reported in the literature in comparable studies or meta-analyses[12,23] with all medication classes having at least 80% of individuals considered as good adherers (adherence >0.8) (Fig. 2). Supplementary Fig. 1 provides an overview of adherence values by showing the relationship between expected and observed consumption across the five medication classes.

For the subset of 185,931 individuals participating in FinnGen, persistence rates, and adherence distributions were consistent with the general population. In the Estonian Biobank, a cohort with similar drug purchase data and genetic information available and 31,074 eligible medication users, we observed generally lower persistence and adherence to statins, clopidogrel, and tamoxifen (Table 1).

### Health and socio-demographic risk factors for persistence and adherence

We studied the effects of the following risk factors, measured at the time of first purchase, on persistence and adherence: age, sex, primary vs secondary prevention, Charlson comorbidity index, years spent in education, living area, mother tongue (a proxy for first-generation immigration status), and receiving any social assistance benefit in the year before starting the treatment. We chose these eight factors because they broadly capture overall health and socio-demographic aspects, were previously examined in the literature[4], and were available from nationwide registers. Basic descriptive statistics for each factor and each medication are reported in Supplementary Data 2. Some socio-demographic factors were consistently associated with lower persistence and adherence across all medications in a

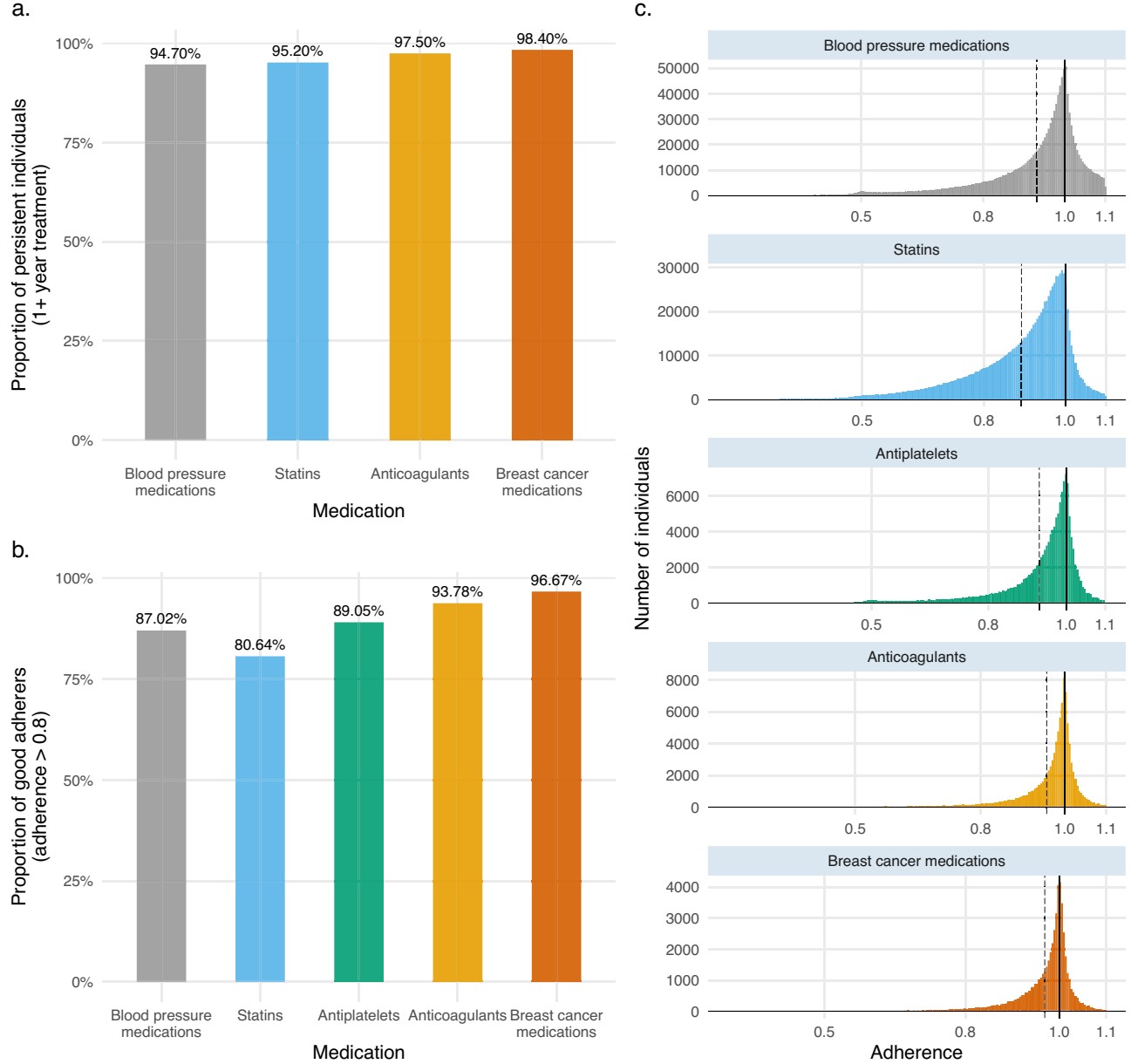

**Fig. 2 | Persistence and adherence to five medications in the general Finnish population. a** Proportion of persistent individuals for each medication class considered in the analysis. **b** Proportion of good adherers (individuals with adherence greater than 0.8) for each medication class. **c** Adherence distribution and mean adherence (dotted line) for the five medication classes considered in the analysis.

multivariable analysis (Fig. 3 considers dichotomized factors while Supplementary Data 3, 4 report similar analysis when continuous factors are treated as such). For example, not speaking Finnish or Swedish and having received social assistance benefits were associated with an average of 3.2% [1.9–4.3%] and 2% [1.1–3.1%] lower adherence across medications, respectively. Starting treatment after a related disease event (secondary prevention) was also consistently associated with higher odds of persistence for statins (odds ratio (OR) = 1.53, 95% confidence interval (CI) = 1.49–1.58, $P = 4.9 \times 10^{-198}$) and increased adherence to statins and antiplatelets (1.7% increase, 95% CI = 1.64–1.76% $P < 2 \times 10^{-308}$ for statins; 0.47% increase, 95% CI = 0.35–0.59%, $P = 7.4 \times 10^{-15}$ for antiplatelets). Age and sex showed discordant effects across different medications after adjustment for the remaining factors. Being female showed a consistent negative effect on persistence for all drugs but showed heterogeneous effects

on adherence. Age also showed heterogeneous effects across different medications (Supplementary Data 3–6). Extending beyond these baseline factors, we further assessed the effect of one or more concurrent treatments (i.e., polytherapy)(Supplementary Data 7, 8). We observed a consistent positive association between having at least one or more treatments and higher adherence and increased odds of persistence (Supplementary Methods).

Overall, the considered health and socio-demographic factors showed a higher effect on persistence than on adherence as indicated by the overall higher variance explained (Supplementary Data 9).

In the Estonian Biobank, when examining variables that are present in both studies, we found consistent results in the direction of effect and statistical significance for both persistence and adherence to statins and age, sex, and years spent in education, but not for secondary prevention (Supplementary Data 10).

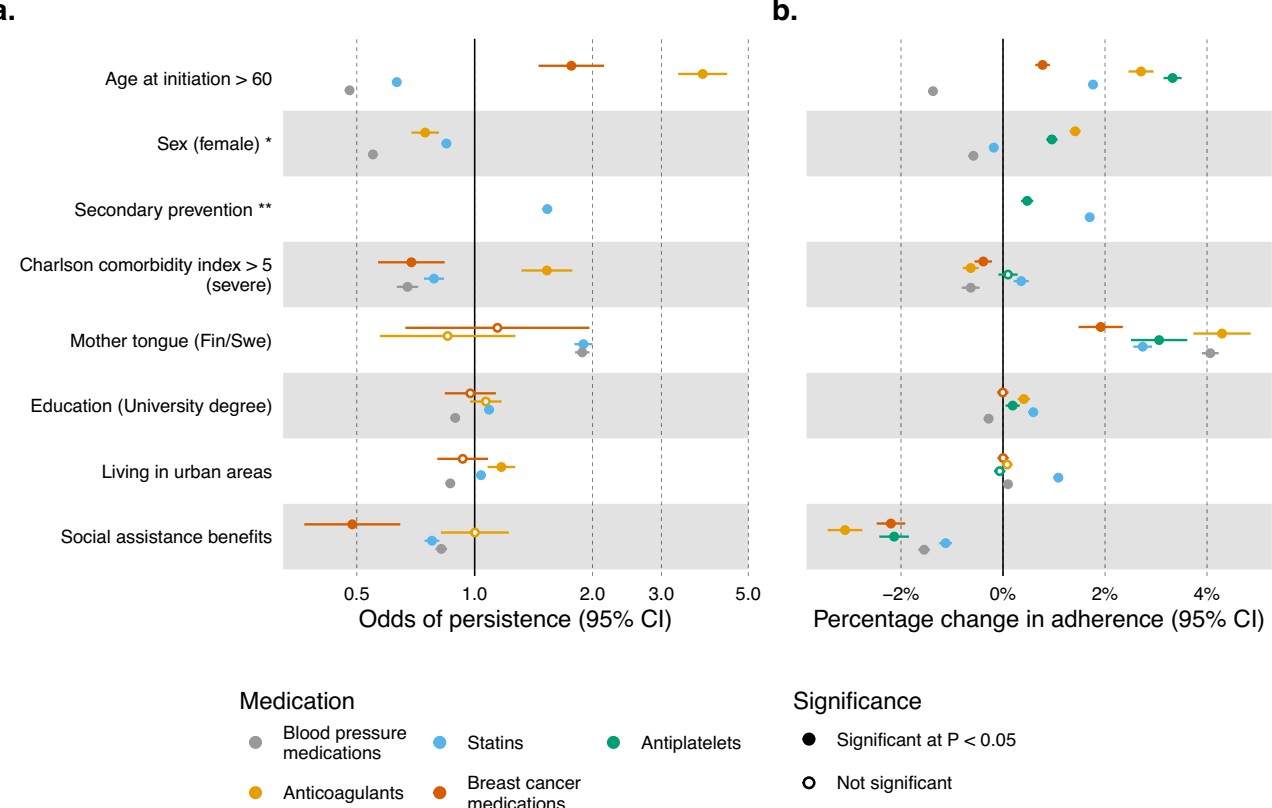

**Fig. 3 | Effect of eight health, demographic, and socioeconomic factors on persistence and adherence in the Finnish population. a** Odds of persistence given each factor's presence. ORs are estimated from a multivariable logistic regression model, error bars represent the 95% confidence interval for the estimates. **b** Percentage change in adherence given each factor's presence. Coefficients are estimated from a multivariable linear regression model and were rescaled to represent the percentage change in adherence given each factor's presence. Error bars represent the 95% confidence interval for the estimate. Descriptive statistics for each risk factor are reported in Supplementary Data 2. In this figure we binarized the continuous risk factors to make effect sizes comparable, however results for non-binarized risk factors are available in Supplementary Data 3, 4. *P* values are two-sided and were calculated by dividing the coefficient values by their standard errors and observing the probability mass corresponding to equal or more extreme values from both tails of the relevant distribution (standard normal distribution for logistic regression and t-distribution for linear regression). No adjustments for multiple comparisons were made. *sex not included for breast cancer medications as all individuals included are females. **secondary prevention is defined only for statins and antiplatelets (Methods).

## Established pharmacogenes marginally associate with persistence but not adherence

We tested the effects of five relevant pharmacogenes on persistence and adherence, in both FinnGen and the Estonian Biobank. We selected the following pharmacogenes based on guidelines from the Clinical Pharmacogenetics Implementation Consortium[24–27], filtering for the drug/gene pairs which were reported with recommended prescribing actions (e.g., alternative therapies or dosing): *ABCG2* for rosuvastatin, *CYP2C9* for fluvastatin, *SLCO1B1* for all statins, *CYP2C19* for clopidogrel, and *CYP2D6* for tamoxifen. For each gene, we defined individual diplotypes and grouped them into phenotypes defined based on each diplotype activity score. We then examined the effect of each phenotype on persistence and adherence. The *CYP2C9* intermediate metabolizer phenotype was nominally significantly associated with lower odds of persistence to fluvastatin in both FinnGen and the Estonian Biobank, and when meta-analyzing the results from both studies ($OR_{meta-analysis} = 0.6$, 95% $CI_{meta-analysis} = 0.43-0.84$, $P_{meta-analysis} = 0.003$) (Fig. 4 and Supplementary Data 11). Similarly, *SLCO1B1* decreased and poor function were associated with lower odds of persistence to all statins ($OR_{meta-analysis} = 0.92$, 95% $CI_{meta-analysis} = 0.86-0.98$, $P_{meta-analysis} = 0.009$; $OR_{meta-analysis} = 0.94$, 95% $CI_{meta-analysis} = 0.88-0.99$, $P_{meta-analysis} = 0.04$, respectively). Decreased *SLCO1B1* function was also significantly associated with lower adherence, although with a neglectable effect (−0.27% change in adherence,

$P = 0.033$) (Fig. 4 and Supplementary Data 12). None of the associations were significant after multiple-testing correction ($P < 0.002$, Bonferroni correction for 19 phenotypes tested).

## Genome-wide association study of drug adherence and persistence

We conducted a GWAS of persistence and adherence to each medication in FinnGen, adjusting for age at initiation, year of birth, sex, genotyping batch, and first ten genetic principal components (Supplementary Fig. 2). No variant showed significant association with either phenotype at a significance level of $P < 1 \times 10^{-8}$ (standard genome-wide significance corrected for multiple-testing across the five medications). Four SNPs, however, were genome-wide significant at the standard $P < 5 \times 10^{-8}$ threshold (Supplementary Data 13). Notably, *rs1339882991* was positively correlated with both persistence and adherence to BP medications and was previously reported to be positively associated with hypertension[28,29]. Further characterization of this and the other genome-wide significant variants is provided in the Supplementary Methods.

Considering the sample size observed for statins ($N = 116,439$), we have at least 80% power to identify genome-wide significant variants with an effect of greater than 2.7% on adherence, assuming a minor allele frequency of 0.3. Such an effect is in line with what is observed for the strongest epidemiological risk factor for statin adherence

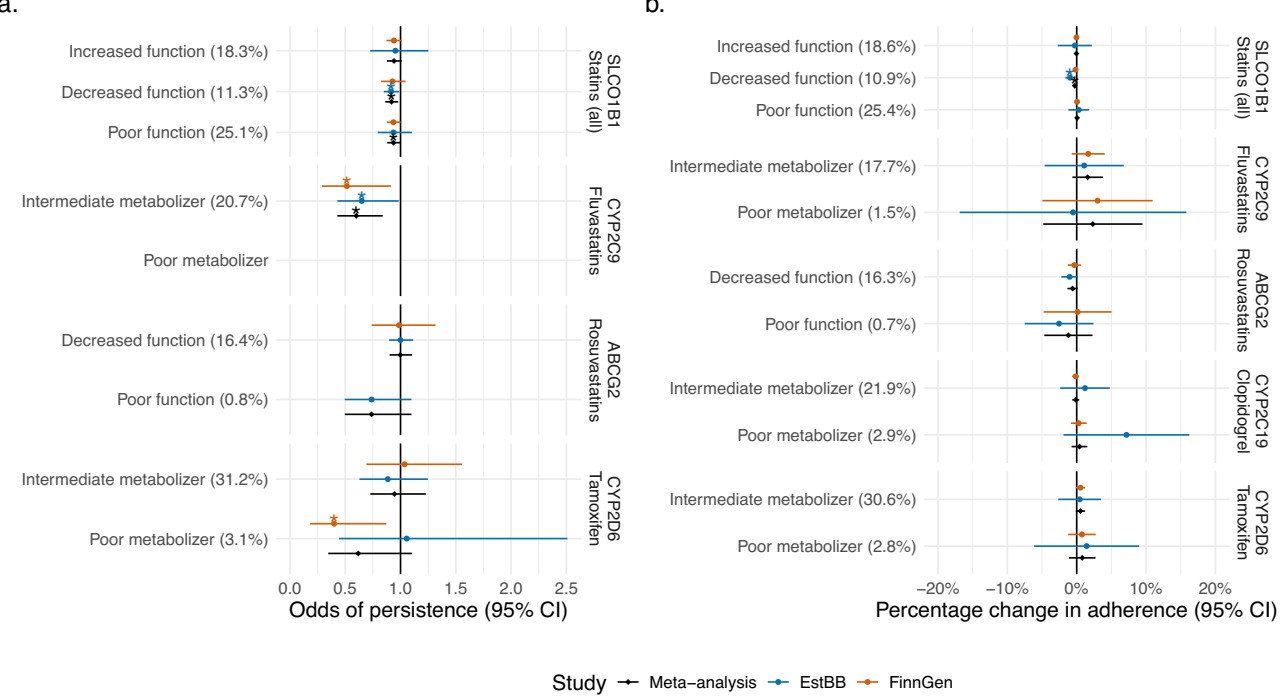

**Fig. 4 | Association between clinically relevant pharmacogene phenotypes and persistence and adherence in FinnGen and the Estonian Biobank. a** Odds of persistence given each phenotype level. ORs are estimated from a logistic regression model, separately for each gene/drug pair ($N = 83,305$. $N$ per each specific gene/drug pair and phenotype level are reported in Supplementary Data 11). **b** Percentage change in adherence given each phenotype level. Coefficients are estimated from a linear regression model, separately for each gene/drug pair ($N = 81,902$. $N$ per each specific gene/drug pair and phenotype level are reported in Supplementary Data 12). Dots represent estimated odds of persistence/percentage change in adherence, error bars represent the 95% confidence interval for the estimate. Stars represent nominal ($P < 0.05$) statistical significance. Estimates are not reported for phenotypes with $n < 5$. For each phenotype, the frequency within each drug user group is reported in parenthesis. $P$ values are two-sided and were calculated by dividing the coefficient values by their standard errors and observing the probability mass corresponding to equal or more extreme values from both tails of the relevant distribution (standard normal distribution for logistic regression and t-distribution for linear regression). No adjustments for multiple comparisons were made.

(Fig. 3). As we expect individual genetic effects for common variants to be smaller than socio-environmental, our results indicate that at the current sample size, the power to identify genetic associations for adherence and persistence is limited.

## Polygenic overlap between persistence, adherence, and other health and behavioral traits

We explored the genetic overlap between persistence and adherence and 33 major diseases, disease risk factors, and behavioral traits (Fig. 5 and Supplementary Data 14, 15). Higher systolic blood pressure was genetically correlated with higher persistence to BP medications ($rg = 0.77$, $P = 3.1 \times 10^{-23}$) and with higher adherence to both statins and BP medications ($rg = 0.17$, $P = 3.4 \times 10^{-06}$ and $rg = 0.4$, $P = 3.5 \times 10^{-30}$, respectively). On the contrary, higher LDL cholesterol was genetically correlated with lower statins adherence ($rg = -0.14$, $P = 0.006$), although when considering a GWAS of LDL cholesterol adjusted by statins usage, the effect was not statistically significant anymore. Notably, both adherence to statins and BP medications were positively genetically correlated with participating in one or more of the optional components of UK Biobank[30] ($rg$ between 0.11 and 0.31, $P < 8.2 \times 10^{-3}$), indicating some shared effect between drug adherence and continued participation in scientific studies.

We wondered if existing polygenic scores (PGS) could be used to predict adherence and persistence and how their effect compared to that of the health and socio-demographic risk factors examined before. Among the 33 PGS examined (Supplementary Figs. 3, 4 and Supplementary Data 16, 17) the strongest effects were observed for systolic blood pressure (SBP) PGS on BP medications persistence

(OR = 1.64 for 1 standard deviation (SD) in PGS, 95% CI = 1.60–1.68, $P < 2 \times 10^{-308}$) and adherence (0.8% increase per 1-SD change in PGS, 95% CI = 0.74–0.86%, $P = 2.6 \times 10^{-126}$); LDL cholesterol (adjusted for statins usage) PGS on statins persistence (OR = 1.24, 95% CI = 1.20–1.28, $P = 6.2 \times 10^{-47}$) and SBP PGS on statins adherence (0.28% increase, 95% CI = 0.22–0.34%, $P = 2 \times 10^{-16}$); coronary artery disease PGS on anti-platelets adherence (0.47% increase, 95% CI = 0.29–0.67%, $P = 4.3 \times 10^{-7}$) and PGS for schizophrenia and adherence to breast cancer medications (0.22% decrease, 95% CI = 0.1–0.34%, $P = 4.2 \times 10^{-4}$). No significant association was observed between any of the PGS and adherence or persistence to DOAC.

In the Estonian Biobank, systolic blood pressure and LDL PGS were also among the strongest effects associated with increased statin adherence, and persistence, respectively (percentage increase in adherence = 0.66%, $P = 1.11 \times 10^{-3}$); (OR = 1.13, 95%CI = 1.093–1.167, $P = 4.65 \times 10^{-9}$)) (Supplementary Data 18, 19).

Overall, these PGSs could explain a lower fraction of the variance in adherence than the combined effect of the health and socio-demographic risk factors examined above, after accounting for age and sex (Supplementary Data 20).

Finally, we wondered whether developing a PGS for adherence would hold clinically meaningful predictive power. We used the GWAS results from FinnGen to build a PGS for statin adherence and tested its association with observed adherence in the Estonian Biobank. We found the PGS to be significantly associated with observed adherence (percentage change in adherence for 1 SD change in PGS = 0.72%, $P = 1 \times 10^{-5}$), however, the small effect size observed suggests such PGS to have low predictive value and clinical relevance.

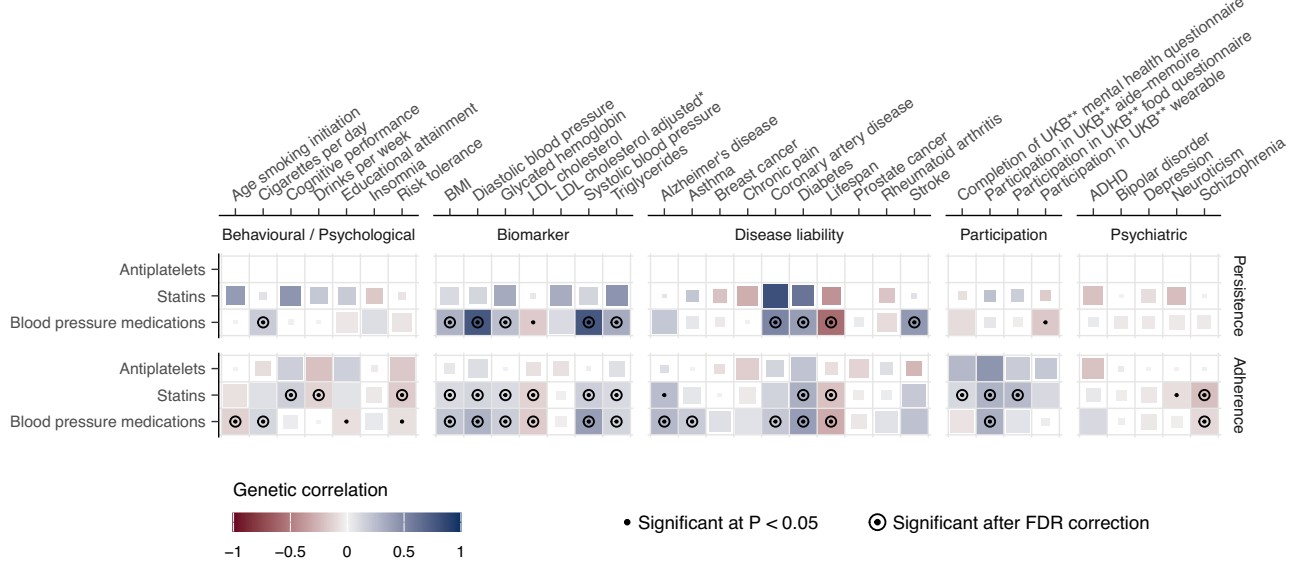

**Fig. 5 | Genetic correlates of persistence and adherence.** Genetic correlations between 33 clinically relevant traits and persistence and adherence. Medications for which genetic correlations could not be computed are not displayed. Blue represents positive correlations, red negative. The size of each tile is proportional to the statistical significance (the bigger the tile, the lower the *P* value). Correlations nominally significant at *P* < 0.05 are marked with a dot, while correlations which were significant after FDR correction (within each medication) are additionally marked with a circle. Two-sided *P* values were calculated using LD score regression. *adjusted by the use of cholesterol-lowering medication. **UKB UK Biobank.

## Sensitivity analysis on persistence definition

Our definition of persistence resulted in higher proportions of persistent individuals compared to what was observed in literature[31]. Thus, as sensitivity analysis, we consider a different definition of persistence, looking at the proportion of individuals that stop medication within one year among those with at least one purchase of the medication. This definition includes a broader population (i.e., does not exclude overbuying behaviors) and considers as not persistence those individuals that stop after two and three purchases, as opposed to only one. With this definition, persistence for statins was 75.6%, consistent with the literature[31]. When looking at the effect of PGS on persistence, we observed directionally consistent effects, but an overall reduction in effect sizes despite the increased number of statistically significant associations (Supplementary Data 21 and Supplementary Fig. 5).

## Discussion

Drug adherence remains a major obstacle to the benefits of pharmacotherapies. Although many external factors (e.g., healthcare- and system-related factors) contribute to drug adherence[4,5], focusing on the patient-specific determinants might help in identifying patients at risk of low adherence. In this study, we provide a comprehensive exploration of patient-specific factors affecting drug persistence and adherence across five different classes of medications, using nationwide high-quality registry data and, for the first time, also exploring genetic factors.

Overall, our results suggest that the need for social assistance and being first-generation immigrants have a moderate to large effect on drug adherence and persistence, which is consistent across medication classes. In comparison, demographic characteristics (age and sex), comorbidity burden, and genetic factors have modest or heterogeneous effects. The only health factor with a robust effect was the presence of a disease related to the medication, which substantially increased adherence among those using drugs for secondary prevention. Similarly, the concurrent use of multiple medications was associated with increased adherence and higher odds of persistence. Although previous literature showed inconsistent[32] or negative[33,34] effects of polypharmacy and adherence, we believe the positive

association in our findings is due to the fact that we are focusing on a restricted set of medications, which generally have higher adherence rates. Furthermore, we speculate that polypharmacy may partially reflect an underlying perception of increased health risk, as it is likely the case also for secondary prevention. In line with previous reports[23], the effects of age and sex were heterogeneous across different medications. Nevertheless, females had consistently lower odds of persistence across all medications. We speculate that a higher rate of adverse drug reactions in women compared to men[35,36], or lower perceived risk for cardiometabolic diseases among women can explain these findings.

We performed three main genetic analyses. First, we evaluated if established pharmacogenes for statins, clopidogrel, and tamoxifen were associated with persistence and adherence to these drugs. Our results indicate that poorer statins metabolism is associated with lower odds of persistence, with a notable effect for *CYP2C9* and fluvastatin. Since increased systemic concentration of statins is associated with a higher risk of developing statin-associated muscle symptoms[25], these findings are in line with the hypothesis of patients discontinuing the treatment as they develop muscle symptoms. However, as these results were consistent in both FinnGen and the Estonian Biobank, but were not significant after multiple-testing correction, larger studies should specifically test this hypothesis. Because pharmacogenomics testing is not widespread in Finland, we do not believe these results are biased by genetically-informed drug prescriptions. Overall, the lack of association between most pharmacogenes and adherence suggests that widespread pharmacogenetic testing before prescribing the medications included in this study is not going to substantially improve adherence in the population, while it could be valuable to tailor the prescribing dose of statins to avoid the risk of developing statin-associated muscle symptoms and improve persistence.

Second, we performed a GWAS of drug adherence and persistence and used these results to understand the overlap with other traits. We did not identify robust individual genetic signals but the identification of four genome-wide significant variants, two of which were not previously reported to be associated with traits underlying the need for medication, warrants for larger genetic studies of

adherence and persistence. Our power calculations indicate that even if this is the largest study of this kind, power is still limited to identify genome-wide significant associations of small effects. For example, at the current sample size, the variants in the pharmacogenes tested, which showed marginal significance in the targeted analysis, would not have been detected at a stringent genome-wide threshold.

A previous study using FinnGen data compared cardiometabolic-related medication user vs non-user or changes in medication use and identified >40 loci reflecting underlying cardiometabolic risk[37]. The differences in the magnitude of genetic signals and genetic correlation profiles in the current study compared to our previous study indicate that drug adherence and persistence have lower genetic contributions and a stronger behavioral component, which does not necessarily reflect the underlying disease risk. In our study, interestingly, we observed genome-wide overlaps with a proxy for study participation, indicating some shared effect between adherence to drug therapy and participation in scientific research, similar to what we recently observed for COVID-19 vaccination uptake[38].

Third, we explored if existing PGS could predict adherence and persistence and found the strongest association between systolic blood pressure PGS and adherence/persistence to BP medications. This association probably reflects individuals with higher BP being more adherent due to higher perceived risk[39]. Finally, while our results could be used to generate a PGS for statin adherence, which significantly replicated in the Estonian Biobank, the variance explained by this PGS was low.

A strength of this study is the use of Finnish registries covering all drug purchases since 1998 which allows an in-depth longitudinal study of drug purchasing. For example, the average treatment length for BP medications was over 10 years. Moreover, by using nationwide data, we could confirm that adherence and persistence measures in FinnGen were similar to what was observed in the general population. Importantly, this study considers both socioeconomic and genetic risk factors, helping to better contextualize the results and bridging the divide between social science and human genetics, as advocated by others[40]. Furthermore, we derived adherence and persistence measures from 5 different medication classes, which allowed us to capture adherence-related factors beyond single disease- and drug-related factors. Finally, both epidemiological and genetic results were replicated in an external study.

A limitation of the study is that our definitions of persistence and adherence can be biased in different ways. First, our persistence definition resulted in higher proportions of persistent individuals compared to what was observed in literature[31]. Nonetheless, we chose this definition to better capture early discontinuation behaviors (i.e., after the first purchase). Moreover, we deemed it fair to compare these individuals only to those included in the adherence analysis, so as not to include overbuying behaviors. Sensitivity analyses suggest that our results are robust to a different definition of persistence and that our stricter inclusion criteria resulted in larger effect sizes at the expense of decreased statistical power.

As for the adherence definition, first we cannot know if individuals are really taking medications and instead rely on the expected and observed consumption. Second, we don't have information on the dose prescribed by the doctor for the entire period and rely on the suggested dosage of the package. For example, individuals which are suggested by the doctor to take half-tablet of statins/day despite package indication, will result in low adherence. Two results indicate that this behavior is not common. First, we do not see enrichment in adherence values that would capture a systematic indication of a non-standard dose of the medication. Second, since 2015, prescription information is available as text. Using regular expressions to extract this information (Methods) we found that in 2019, 96% of the prescriptions were indicating 1 statin tablet/day. Finally, the observed adherence is higher than what was reported in a previous Finnish study[41]. Considering only individuals that had taken medication for at least 12 months and excluding long inactive periods might have contributed to higher adherence values. Nonetheless, early discontinuation is captured by our persistence measure, while long breaks in drug usage might reflect hospital admission, partial emigration, or other external factors. Overall, we used a definition of adherence that is more robust to large external shocks and better suited to capture smaller and consistent variations in drug purchases over longer time periods.

In conclusion, our results suggest a limited role of genetic risk factors, including clinically relevant pharmacogenes, in explaining persistence and adherence for five common medication classes that require long-term, regular therapy. While larger GWASs are needed to identify smaller genetic effects, the observed genetic overlap with behavioral traits, might complicate the interpretation of the results. The robust associations with measures of social deprivation suggest that we need a better understanding of the drivers of poor adherence in these socio-economically disadvantaged groups. Identifying these drivers might offer points of intervention in clinical practice, or help recognize system-wide barriers for drug adherence.

## Methods

### FinRegisty and FinnGen study population

The FinRegistry dataset[20] contains data from Finnish nationwide health, demographic, and socioeconomic registries, for a total of 7,166,416 individuals. To ensure completeness of information for the variables considered in the epidemiological analysis, we included only individuals who were residents of Finland and alive on 1 January 2010 (5,339,804 individuals in total). The FinnGen dataset[21] (Supplementary Methods) additionally contains genetic data for a subset of the Finnish population ($N = 430,885$ for data freeze 10). We studied the two drug usage phenotypes in adults only, meaning individuals needed to be at least 18 years old at treatment initiation.

### Kela drug purchase register

The Social Insurance Institution of Finland (Kela) is a government agency that provides basic economic security by financial support for everyone living in Finland. The Kela drug purchase register (available both in the FinRegistry and FinnGen datasets) contains information about any reimbursable prescription medication bought from pharmacies in Finland between 1995 and 2020. We included only purchases from 1998 on to ensure better homogeneity with the data from the other registries. Each data entry contains information about the date of purchase, the ATC code of the medication, the number of purchased packages, and the Nordic Article Number (VNR) associated with the package, which gives information about the quantity and dosage of the medication. When the information was available, we additionally excluded those purchases made using pharmacies dose distribution services, which regularly provides patients with 2-week worth of medications at a time.

### The Estonian Biobank study population

The Estonian Biobank (EstBB) (Supplementary Methods) is a volunteer-based biobank that has a sample size of approximately 211,000 participants, comprising over 20% of the adult population of Estonia[22]. The EstBB is linked with the National Health Insurance Fund's (NHIF) database, which was used as the EHR source. NHIF covers over 95% of the Estonian population as a result of universal health care in Estonia and thus provides information on diagnoses, prescribed and dispensed drugs for up to 19 years (2004–2023) from primary and specialist care. Information on prescribed and dispensed drugs is collected from all pharmacies in Estonia. The NHIF database contains information on the prescribed and dispensed drug name, date of prescription and dispensing, International Classification of Diseases (ICD-10) diagnosis codes, anatomical-therapeutic-chemical (ATC) code of the drug, number of packs dispensed, number of tablets in

each package, and concentration of active ingredient in each tablet. The study population in the Estonian Biobank cohort consisted of individuals who initiated their treatment between 2007 and 2023.

## Medications included

We included medications commonly prescribed and purchased, that are typically used for a prolonged time and do not have a significant direct effect on wellbeing. The classes of medication we included are as follows: statins (C10AA*−all types of statins without differentiation), blood pressure medications (C02*− antihypertensives, C03*−diuretics, C08*−calcium channels blockers, C09*−agents acting on the renin-angiotensin system), antiplatelets (B01AC04−clopidogrel and B01AC30−acetylsalicylic acid in combination with dipyridamole), breast cancer medications (L02BA01−tamoxifen, L02BG03−anastrozole, L02BG04−letrozole, L02BG06 - exemestane), direct oral anticoagulants (DOAC) (B01AF01−rivaroxaban, B01AF02−apixaban, B01AF03−edoxaban, B01AE07−dabigatran etexilate). For breast cancer medications, we included only females, starting the treatment after a breast cancer diagnosis. For DOAC, only patients starting the treatment after an atrial fibrillation diagnosis were included.

The following drug classes or drugs were considered in the Estonian Biobank: statins drug class (ATC C10AA*), and separately clopidogrel (ATC B01AC04), tamoxifen (L02BA01), fluvastatin (ATC C10AA04), and rosuvastatin (ATC C10AA07). The patients were included if they were over the age of 18 at the start of treatment initiation.

## Persistence

We define persistence[18] as a binary phenotype, comparing individuals taking the medication for at least 12 months (i.e., all individuals included in the adherence analysis) versus patients early-discontinuing the drug after one purchase. To avoid right censoring, we considered early discontinuation only purchases made at least 2 years before the end of follow-up. The end of the follow-up date was either death date, moving abroad date or the last date for which the data were available (01.01.2020).

In the Estonian Biobank, for the non-persistence phenotype, we included individuals discontinuing the treatment after one purchase that contained less than three packages and 100 tablets. We limited the drug supply that could be bought with one purchase to exclude individuals who may have stocked medications due to moving abroad or due to other reasons. We considered as early discontinuation only purchases made before 01.01.2022 (the last date for which the data were available (28.03.2023)) and at least 1 year before the death date.

## Adherence

For each individual, we considered all registered purchases for each class of medication and reconstructed individual drug purchasing trajectories. We then estimated adherence using the medication possession ratio measure (MPR)[19]:

$$MPR = \frac{days\ of\ supply\ during\ observation\ period}{days\ in\ observation\ period} \quad (1)$$

where
- days of supply during the observation period is the total number of tablets purchased in the trajectory, normalized assuming drug-specific daily doses (e.g., statins: 1 tablet/day).
- days in the observation period is the sum of the intervals (in days) between all purchases considered in the trajectory.

When reconstructing the trajectories, we excluded the last purchase and purchases for which the interval to the next one was at least 150 days without tablets available. We considered this as a break in the treatment, which should not be taken into account in the adherence

calculation. When patients were switching to different formulations (within the same medication class), and the interval between purchases was shorter than the drug supply, we discarded the leftover tablets of the previous formulation. We included only individuals with at least 1 year of purchases. We further excluded individuals with adherence greater than 1.1, to exclude overbuying behaviors.

In the Estonian Biobank, patients were included if they had purchased at least 150 tablets. Individual purchase trajectories were constructed and adherence was estimated using the same criteria specified above with the exception of excluding 150 day intervals without available treatment. The Estonian medical system does not offer services where the treatment would be provided by the hospital without it being recorded in the drug purchase database (with the exception of inpatient care). Therefore, we considered all gaps in drug purchases as true breaks in treatment.

## Sex and gender

Sex was used as a covariate in all of the analyses and no analyses were run separately in each sex. In FinRegistry, sex was defined as recorded in the population registries from the Digital and Population Data Services Agency. In FinnGen, genetically determined sex was used, after quality control removal for individuals with discrepancy between sex reported in the population registries and genetically determined sex. In the Estonian Biobank, sex was extracted from the Estonian National Identity number, which is created based on the sex recorded in the Estonian Birth Registry. Quality assurance was carried out where only individuals whose sex from the National Identity number matched the sex in genotype data were included.

## Health, demographic, and socioeconomic risk factors for persistence and adherence

We assessed the effect of the following eight epidemiological risk factors on persistence and adherence:
- sex (not considered for breast cancer medications).
- age at treatment initiation: age when the first purchase was recorded.
- secondary prevention: whether the treatment was started after a major event related to the medication considered. The following diagnosis (ICD-10 codes) were considered:

  - statins: major coronary heart disease (I20.0, I21, I22), cerebrovascular diseases (I60-I69), atherosclerosis (excluding cerebral, coronary, and PAD) (I70).
  - antiplatelets: stroke (I60−I64), transient ischemic attack (G45), myocardial infarction (I21, I22).
  - BP medications: no specific outcome was considered, as this is a broader class of medications.
  - breast cancer medications: breast cancer (C50)−only secondary prevention patients were included.
  - DOAC: atrial fibrillation (I48), pulmonary embolism or venous thromboembolism (I26, I80, I87.1)−only secondary prevention patients were included.

- Charlson comorbidity index (at treatment initiation): we considered patients comorbidity status at treatment initiation by computing the Charlson comorbidity index (CCI) based on all the diagnoses registered previous to the first purchase. This was computed using the R package ICCI, which uses the R package comorbidity (https://cran.r-project.org/web/packages/comorbidity/index.html).
- years of education (at treatment initiation): years expected to complete the recorded education level. When the information at treatment baseline was not available, the first available record was considered.

- living area (at treatment initiation): urban *vs* rural, as classified by the Finnish digital and population data agency (DVV).
- Mother tongue: Finnish or Swedish (the two official languages of Finland) vs any other language.
- social assistance benefits: receiving (or not) any amount of social assistance benefit in the year before treatment initiation.

The effect sizes of each factor on persistence and adherence were derived by using a multivariate logistic/linear regression model including all the factors, adjusting for year of birth.

For persistence, the estimated effects are reported as OR from the logistic regression model, while for adherence, we reported the percentage change in adherence by rescaling the log(OR) from the linear model by a factor of 1.1 (maximum adherence value). To provide a better interpretation of the results, in Fig. 3, we report the effect sizes for a model where continuous variables were dichotomized according to the following thresholds: age at initiation >60; CCI >5; education, university degree, or higher. The same effect sizes keeping the continuous variables as such (standardized to have mean 0 and variance equal to 1) are reported in Supplementary Data 4, 5.

Among the health, demographic, and socioeconomic factors considered in FinRegistry, the following were available in the Estonian Biobank: sex, age at treatment initiation, birth year, type of prevention, and years of education. Age at treatment initiation, birth year, and years of education were standardized to have a mean of 0 and a variance equal to 1. We assessed the association between statin persistence and adherence with these factors by using multivariate logistic and linear regression models, respectively, while only considering individuals with complete information for all variables (N = 27,339).

## Pharmacogenes selection and phenotypes definition

We selected the pharmacogenes to test based on the guidelines provided by the Clinical Pharmacogenetics Implementation Consortium[24]. The CPIC assigns levels to gene/drug pairs based on PharmGKB[42] annotation levels, recommendations by external groups (e.g., FDA, EMA) or by other professional societies. We selected gene/drug pairs to match all the ATC codes included in our analyses, and with a CPIC level A or B, meaning that prescribing actions (e.g., alternative therapies or dosing) are recommended when the pharmacogene is present. The gene/drug pairs we selected are: *ABCG2* for rosuvastatin, *CYP2C9* for fluvastatin, *SLCO1B1* for all statins, *CYP2C19* for clopidogrel, and *CYP2D6* for tamoxifen.

For each gene and for each individual included in the analyses, we defined their star alleles diplotypes to test starting from the imputed phased genotypes available in FinnGen and using Stargazer[43,44]. Having genotype data available, we could define only diplotypes captured by SNP and not by structural variations (gene deletions or copy number variations). We then grouped individuals based on the phenotypes also provided as output by Stargazer and reflecting the metabolic activity score associated with each specific diplotype (poor/decreased/increased function for *SLCO1B1* and *ABCG2*, poor/intermediate/ultra-rapid metabolizer for *CYP2D6*, *CYP2C9*, and *CYP2C19*). In the Estonian Biobank star alleles were called from imputed phased genotypes available using PharmCAT[45] and an in-house pipeline[46].

We used a logistic/linear regression model to estimate the association of each phenotype with persistence/adherence, adjusting by sex, age at initiation (standardized), and first ten principal genetic components (standardized). We only tested phenotypes when defined for at least five individuals in the adherence analysis, or for at least five persistent/non-persistent individuals in the persistence analysis.

## Genetic risk factors of persistence and adherence

We used REGENIE[47] to run a GWAS of persistence and adherence to each medication (see Supplementary Methods for details about genotyping and imputation in FinnGen). Persistence was treated as a

binary phenotype, while adherence was kept as a continuous trait, but scaled so to have mean 0 and variance equal to 1. For both analyses, we adjusted for sex, genotyping batch, first ten principal genetic components, age at first purchase, and year of birth. The resulting summary statistics were post-processed to keep only SNPs with an INFO score ≥0.8 and MAF ≥1%.

Genetic correlations between persistence and adherence and 33 publicly available traits (listed in Supplementary Data 14) were calculated using linkage disequilibrium score regression[48].

PGSs for the same traits were derived using PRS-CS[49]. Their association with persistence and adherence was assessed by running a logistic/linear regression model for each PGS individually, adjusting for sex, age at initiation, and the first ten genetic principal components (standardized).

Weights for the same traits (Supplementary Data 14) were applied to the Estonian Biobank individuals using PLINK[50]. Similarly, the association of the PGSs with statin persistence and adherence was assessed by running a logistic or linear regression model for each PGS individually, adjusting for sex, age at initiation, and the first ten principal components. The PGS score, age at treatment initiation, and principal components were standardized.

We also derived a PGS for adherence to statins from the summary statistics of the GWAS in FinnGen using PRS-CS and applied it to the Estonian Biobank individuals using PLINK. We then tested the association between statin adherence PGS and the adherence observed in the Estonian Biobank using a linear regression model, adjusting for sex, age at initiation, and the first ten genetic principal components.

## Ethics declarations

FinRegistry is a collaboration project of the Finnish Institute for Health and Welfare (THL) and the Data Science Genetic Epidemiology research group at the Institute for Molecular Medicine Finland (FIMM), University of Helsinki. The FinRegistry project has received the following approvals for data access from the National Institute of Health and Welfare (THL/1776/6.02.00/2019 and subsequent amendments), DVV (VRK/5722/2019-2), Finnish Center for Pension (ETK/SUTI 22003), and Statistics Finland (TK-53-1451-19). The FinRegistry project has received IRB approval from the National Institute of Health and Welfare (Kokous 7/2019).

Patients and control subjects in FinnGen provided informed consent for biobank research, based on the Finnish Biobank Act. Alternatively, separate research cohorts, collected prior to the Finnish Biobank Act came into effect (in September 2013) and the start of FinnGen (August 2017), were collected based on study-specific consents and later transferred to the Finnish biobanks after approval by Fimea (Finnish Medicines Agency), the National Supervisory Authority for Welfare and Health. Recruitment protocols followed the biobank protocols approved by Fimea. The Coordinating Ethics Committee of the Hospital District of Helsinki and Uusimaa (HUS) statement number for the FinnGen study is Nr HUS/990/2017.

The FinnGen study is approved by Finnish Institute for Health and Welfare (permit numbers: THL/2031/6.02.00/2017, THL/1101/5.05.00/ 2017, THL/341/6.02.00/2018, THL/2222/6.02.00/2018, THL/283/ 6.02.00/2019, THL/1721/5.05.00/2019, and THL/1524/5.05.00/2020), Digital and population data service agency (permit numbers: VRK43431/2017-3, VRK/6909/2018-3, and VRK/4415/2019-3), the Social Insurance Institution (permit numbers: KELA 58/522/2017, KELA 131/ 522/2018, KELA 70/522/2019, KELA 98/522/2019, KELA 134/522/2019, KELA 138/522/2019, KELA 2/522/2020, KELA 16/522/2020), Findata permit numbers THL/2364/14.02/2020, THL/4055/14.06.00/2020, THL/3433/14.06.00/2020, THL/4432/14.06/2020, THL/5189/14.06/ 2020, THL/5894/14.06.00/2020, THL/6619/14.06.00/2020, THL/209/ 14.06.00/2021, THL/688/14.06.00/2021, THL/1284/14.06.00/2021, THL/ 1965/14.06.00/2021, THL/5546/14.02.00/2020, THL/2658/14.06.00/ 2021, THL/4235/14.06.00/2021, Statistics Finland (permit numbers: TK-

53-1041-17 and TK/143/07.03.00/2020 (earlier TK-53-90-20) TK/1735/07.03.00/2021, TK/3112/07.03.00/2021) and Finnish Registry for Kidney Diseases permission/extract from the meeting minutes on 4 July 2019.

The Biobank Access Decisions for FinnGen samples and data utilized in FinnGen Data Freeze 10 include: THL Biobank BB2017_55, BB2017_111, BB2018_19, BB_2018_34, BB_2018_67, BB2018_71, BB2019_7, BB2019_8, BB2019_26, BB2020_1, BB2021_65, Finnish Red Cross Blood Service Biobank 7.12.2017, Helsinki Biobank HUS/359/2017, HUS/248/2020, HUS/150/2022 § 12, §13, §14, §15, §16, §17, §18, and §23, Auria Biobank AB17-5154 and amendment #1 (August 17 2020) and amendments BB_2021-0140, BB_2021-0156 (August 26 2021, Feb 2 2022), BB_2021-0169, BB_2021-0179, BB_2021-0161, AB20-5926 and amendment #1 (April 23 2020)and it´s modification (Sep 22 2021), Biobank Borealis of Northern Finland_2017_1013, 2021_5010, 2021_5018, 2021_5015, 2021_5023, 2021_5017, 2022_6001, Biobank of Eastern Finland 1186/2018 and amendment 22 § /2020, 53§/2021, 13§/2022, 14§/2022, 15§/2022, Finnish Clinical Biobank Tampere MH0004 and amendments (21.02.2020 & 06.10.2020), §8/2021, §9/2022, §10/2022, §12/2022, §20/2022, §21/2022, §22/2022, §23/2022, Central Finland Biobank 1-2017, and Terveystalo Biobank STB 2018001 and amendment 25th Aug 2020, Finnish Hematological Registry and Clinical Biobank decision 18 June 2021, Arctic biobank P0844: ARC_2021_1001.

The analysis of individual-level data from the Estonian Biobank was carried out under ethical approval nr 1.1-12/624 from the Estonian Committee on Bioethics and Human Research (Estonian Ministry of Social Affairs), using data according to data release application nr T12 6−7/GI/11086 from the Estonian Biobank.

## Reporting summary

Further information on research design is available in the Nature Portfolio Reporting Summary linked to this article.

## Data availability

Data dictionaries for FinRegistry are publicly available on the FinRegistry website (www.finregistry.fi/finnish-registry-data). Access to the FinRegistry data can be obtained by submitting a data permit application for individual-level data to the Finnish social and health data permit authority, Findata (https://asiointi.findata.fi/). The application includes information on the purpose of data use; the requested data, including the variables, definitions of the target and control groups, and external datasets to be combined with FinRegistry data; the dates of the data needed; and a data utilization plan. The requests are evaluated case by case. Once approved, the data are sent to a secure computing environment (Kapseli) and can be accessed within the European Economic Area and within countries with an adequacy decision from the European Commission. The Finnish biobank data can be accessed through the Fingenious services (https://site.fingenious.fi/en/) managed by FINBB. Access to individual-level data from the Estonian Biobank can be obtained through the Estonian Biobank, following standard data access procedures (https://genomics.ut.ee/en/content/estonian-biobank). All research using the Estonian Biobank data is regulated by the Human Genes Research Act and must be approved by the Estonian Committee of Bioethics and Human Research. Summary statistics for the GWAS of adherence and persistence to statins and blood pressure medications are available in the GWAS catalog under accession codes GCST90448318, GCST90448319, GCST90448320, GCST90448321. All other data supporting the findings described in this manuscript are available in the article and its Supplementary Information files.

## Code availability

The analysis code used to produce the results is available on GitHub at: https://github.com/dsgelab/drugs-persistence-adherence.

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

## Acknowledgements

We are grateful to Finnish individuals, whose data made this study possible. We would also like to thank the entire FinRegistry team for making the data available for the study. We want to acknowledge the participants and investigators of the FinnGen study. The FinnGen project is funded by two grants from Business Finland (HUS 4685/31/2016 and UH 4386/31/2016) and the following industry partners: AbbVie Inc., AstraZeneca UK Ltd, Biogen MA Inc., Bristol Myers Squibb (and Celgene Corporation & Celgene International II Sàrl), Genentech Inc., Merck Sharp & Dohme LCC, Pfizer Inc., GlaxoSmithKline Intellectual Property Development Ltd., Sanofi US Services Inc., Maze Therapeutics Inc., Janssen Biotech Inc, Novartis AG, and Boehringer Ingelheim International GmbH. Following biobanks are acknowledged for delivering biobank samples to FinnGen: Auria Biobank (www.auria.fi/biopankki), THL Biobank (www.thl.fi/biobank), Helsinki Biobank (www.helsinginbiopankki.fi), Biobank Borealis of Northern Finland (https://www.ppshp.fi/Tutkimus-ja-opetus/Biopankki/Pages/Biobank-Borealis-briefly-in-English.aspx), Finnish Clinical Biobank Tampere (www.tays.fi/en-US/Research_and_development/Finnish_Clinical_Biobank_Tampere), Biobank of Eastern Finland (www.ita-suomenbiopankki.fi/en), Central Finland Biobank (www.ksshp.fi/fi-FI/Potilaalle/Biopankki), Finnish Red Cross Blood Service Biobank (www.veripalvelu.fi/verenluovutus/biopankkitoiminta), Terveystalo Biobank (www.terveystalo.com/fi/Yritystietoa/Terveystalo-Biopankki/Biopankki/) and Arctic Biobank (https://www.oulu.fi/en/university/faculties-and-units/faculty-medicine/northern-finland-birth-cohorts-and-arctic-biobank). All Finnish Biobanks are members of BBMRI.fi infrastructure (www.bbmri.fi). Finnish Biobank Cooperative -FINBB (https://finbb.fi/) is the coordinator of BBMRI-ERIC operations in Finland. The Finnish biobank data can be accessed through the Fingenious® services (https://site.fingenious.fi/en/) managed by FINBB. The authors for the Estonian Biobank would like to express our gratitude to all EstBB participants who made it possible to conduct this study. Data analysis was carried out in part in the High-Performance Computing Centre of University of Tartu. This work was written at writing retreats and writing days organized by the University of Tartu Institute of Genomics. The Estonian Biobank Research Team responsible for data collection, genotyping, QC and imputation: Andres Metspalu, Tõnu Esko, Reedik Mägi, Mari Nelis and Georgi Hudjashov. This study has received funding from the European Union's Horizon 2020 research and innovation program under grant agreement No 101016775; from the European Research Council (ERC) under the European Union's Horizon 2020 research and innovation program (grant number 945733); from Academy of Finland fellowship grant N. 323116; from the European Union through the European Regional Development Fund Project No. 2014-2020.4.01.15-0012 GENTRANSMED; the European Union's Horizon 2020 Research and Innovation Program under Grant agreement 847776 and 964874; the Estonian Research Council grantsPSG615 and PRG184.

## Author contributions

Study design: M.C., A.C., S.J., M.N., K.L., L.M., and A.G. Data analysis: M.C., A.C., and H.M.K. Results interpretation: M.C., A.C., H.M.K., S.J., P.V., T.K., M.F., S.R., M.P., M.N., K.L., L.M., and A.G. Writing— original draft: M.C., A.C., and A.G. All authors were involved in further drafts of the manuscript and revised it critically for content. All authors gave final approval of the version to be published. M.C., A.C., and H.M.K. contributed equally to this work.

## Competing interests

M.C. became an employee of Nightingale Health Plc after this work was completed. A.G. is the CEO and founder of Real World Genetics Oy. The remaining authors declare no competing interests.

## Additional information

## FinnGen

Markus Perola[5], Samuli Ripatti[1,9,10] & Andrea Ganna[1,11,12]

## Estonian Biobank Research Team

Andres Metspalu[4], Lili Milani[4], Tõnu Esko[4], Reedik Mägi[4], Mari Nelis[4] & Georgi Hudjashov[4]

