## [Transparent Peer Review file · Nature Communications]

Socio-demographic and genetic risk factors for drug adherence and persistence across 5 common medication classes

Corresponding Author: Professor Andrea Ganna

Version 0:

Reviewer comments:

Reviewer #1

(Remarks to the Author)

Using high-quality nationwide health registry data and genome wide genotyping, the authors evaluated the impact of socio-demographic and genetic risk factors on adherence and persistence for 5 common medication classes. This is an interesting study examining an important problem. I have several questions as follows:

1. I wonder the validity of using polygenic scores (PGS) for certain diseases or behavioral traits to predict drug adherence. As the authors showed in Suppl Table 15, the variance explained by PGS is extremely low. Despite the challenges, there is potential for PGS to contribute to personalized medicine, including personalized strategies for improving drug adherence. For example, understanding a patient's genetic predisposition to certain side effects might help in choosing the most suitable medication, thereby improving adherence. I wonder whether the authors should direct develop and examine the PGS using drug adherence as a trait instead of using existing PGS for certain diseases; whether this PRS help predicting drug adherence, in addition to socio-economic.
2. Most PRS have historically been conducted on populations of European descent. This bias may limit the applicability and accuracy of PRS in non-European populations. Does this affect the analyses in this study?
3. Minor: Table 1 in Supplementary PDF has some formatting problem.

Reviewer #2

(Remarks to the Author)

This is a very clearly written paper describing a fairly large number of important analyses related to adherence to five commonly prescribed medication classes. The genetic results are especially important even though they suggest only modest contributions to adherence/persistence. The conclusions are well supported by the results. I have only minor comments overall:

This statement in the introduction is not entirely accurate: "Nonetheless, the impact of clinically relevant pharmacogenetic variants on drug adherence has not yet been studied." Perhaps the "clinically relevant" qualifier makes it less so but a quick Pubmed search showed several genetic studies of drug adherence. A few examples:

<https://www.mdpi.com/1422-0067/24/6/5636>

<https://academic.oup.com/cardiovasces/article/114/8/1073/5033445>

<https://journals.plos.org/plosone/articleid=10.1371/journal.pone.0029186>

This statement in the introduction has significant implications for the interpretation of this work: "We considered five commonly prescribed medications that are typically used for long-term, continuous, regular therapy, and do not have directly observable effects on symptoms" and warrants further discussion, especially as it relates to side effects, which may have an equally if not greater effect on adherence, especially in terms of genetic effects. Doesn't uncontrolled, elevated blood pressure in particular have a noticeable effect to the patient? Wouldn't the same be true of a patient who develops an atrial

fibrillation or embolism due to Warfarin non-adherence? Is the statement even relevant to the goals of the study? If not, it could be removed.

The SNPs/genes associated with the three nearly GWS hits that did not appear in Open Target Genetics still warrant discussion in both the Results and Discussion.

Additional bioinformatics analysis of the top associations would improve the paper. Do the SNPs have regulatory potential, are they eQTLs/mQTLs, transcription factor binding sites, etc.

Reviewer #3

(Remarks to the Author)

The authors have undertaken a large-scale study using FinnGen and the Estonian biobank to evaluate factors (social, clinical and genetic) associated with poor adherence and persistence to medicines. They found that need for social assistance and immigration status showed a negative effect on persistence and adherence – this has been identified before, and therefore not surprising. Investigation of genomic factors did not identify any meaningful associations, which is also not surprising.

Although the authors have done a lot of work on this paper, and the paper is well written, the only positive associations identified are not novel. Furthermore, the genomic investigations are unsurprisingly negative and probably reflect the tenuous underlying hypotheses.

As the authors acknowledge, just because somebody picks up their medicines from a pharmacy does not necessarily mean they are taking them. All studies have shown that adherence deteriorates the more medicines the person is supposed to be taking – this gets worse when you have to take the medicines more than once per day. Hence a major omission from the study is the lack of any assessment of the number of medicines, and how often per day, being taken by each individual which should be an important indicator of adherence.

I am also confused by the medications studied:

- Antihypertensives are classified separately from diuretics, calcium channel blockers, RA acting drugs (page 20, lines 442-443). What were the other antihypertensives?
- Why wasn't acetylsalicylic acid considered on its own? Seems to have been considered together with dipyridamole.
- Caplacizumab is not an anticoagulant.

Also tolerability varies within each of these classes, and therefore classifying drugs as a class is likely to miss the effects of individual drugs on tolerability. Another major issue, highlighted by the authors, is inaccurate assessment of doses.

Version 1:

Reviewer comments:

Reviewer #1

(Remarks to the Author)

The authors have addressed all my comments. I have no further questions.

Reviewer #2

(Remarks to the Author)

The authors fully addressed my minimal concerns with the paper and made appropriate changes to the text.

Reviewer #3

(Remarks to the Author)

The authors have made most of the changes following the last review. However, an important aspect NOT answered is the effect of the number of medicines taken per day on adherence (reviewer #3, comments 1,2). The applicants may want to look at these papers, and ideally include an assessment of medication burden on non-adherence in their paper.

Zelko E, Klemenc-Ketis Z, Tusek-Bunc K. Medication adherence in elderly with polypharmacy living at home: a systematic review of existing studies. *Mater Sociomed* 2016;28:129–32.

Foley L, Larkin J, Lombard-Vance R, et al. Prevalence and predictors of medication non-adherence among people living with multimorbidity: a systematic review and meta-analysis. *BMJ Open* 2021;11:e044987.

Version 2:

Reviewer comments:

Reviewer #3

(Remarks to the Author)

The authors have undertaken a further analysis on polypharmacy and adherence. They have included a couple of lines in the main paper, and included supplementary details.

The authors find a positive "association between between having at least one or more treatment and higher adherence and increased odds of persistence".

However, there is no discussion of this finding especially since it is discordant with other literature on adherence which shows that adherence worsens as the person has to take an increasing number of medications. Why do the authors think their results provide an effect which is discordant with other literature?

Response to the reviewers:

We would like to thank the reviewers for their thoughtful comments and suggestions. We have thoroughly addressed the issues raised by the reviewers. Our responses to the reviewers' comments are colored in blue in the following, while the new texts added to the manuscript are colored in red.

Reviewer #1:

Using high-quality nationwide health registry data and genome wide genotyping, the authors evaluated the impact of socio-demographic and genetic risk factors on adherence and persistence for 5 common medication classes. This is an interesting study examining an important problem. I have several questions as follows:

Comment 1

I wonder the validity of using polygenic scores (PGS) for certain diseases or behavioral traits to predict drug adherence. As the authors showed in Suppl Table 15, the variance explained by PGS is extremely low. Despite the challenges, there is potential for PGS to contribute to personalized medicine, including personalized strategies for improving drug adherence. For example, understanding a patient's genetic predisposition to certain side effects might help in choosing the most suitable medication, thereby improving adherence. I wonder whether the authors should direct develop and examine the PGS using drug adherence as a trait instead of using existing PGS for certain diseases; whether this PRS help predicting drug adherence, in addition to socio-economic.

Response

We thank the reviewer for the comment and agree that it would indeed be valuable to develop a specific polygenic score for drug adherence. However, based on our findings such a score would likely be poorly predictive of drug adherence. Firstly, we observed no genome-wide significant effects on drug persistence and adherence, leading to non-significant or very low SNP-heritability for these traits (significant heritability was observed only for adherence to statins and BP medications: $h^2_{statins} = 0.037$, $P_{statins} = 2.1 \times 10^{-13}$; $h^2_{BP\ medications} = 0.036$, $P_{BP\ medications} = 1.3 \times 10^{-14}$). Secondly, when we applied a PGS for adherence to statins developed in FinnGen on the data from Estonian Biobank, for replication purposes, despite observing a positive significant association, suggesting that our phenotype definitions indeed captures adherence and holds across cohorts, the effect size of such association was relatively small (change in adherence for 1 SD change in PGS = 0.72%, $P = 1 \times 10^{-5}$), suggesting a low value of a PGS for drug adherence in predicting it. We found this to be a valuable suggestion and we changed the manuscript at line 265 to emphasize this point more:

Finally, we wondered whether developing a PGS for adherence would hold clinically meaningful predictive power. We used the GWAS results from FinnGen to build a PGS for statin adherence and tested its association with observed adherence in Estonian Biobank. We found the PGS to

be significantly associated with observed adherence (percentage change in adherence for 1 SD change in PGS = 0.72%, $P = 1 \times 10^{-5}$), however the small effect size observed suggests such PGS to have low predictive value.

Comment 2

Most PRS have historically been conducted on populations of European descent. This bias may limit the applicability and accuracy of PRS in non-European populations. Does this affect the analyses in this study?

Response

We thank the reviewer for pointing this out and agree this is a valid concern when applying PGS across different ancestries. In this study however this factor does not limit the applicability of PGS as Finnish individuals are considered of European ancestry and PGSs have been proven to transfer well in FinnGen (as shown in Mars et al. "Genome-wide risk prediction of common diseases across ancestries in one million people." Cell genomics. 2022 - <https://doi.org/10.1016/j.xgen.2022.100118>)

Comment 3

Minor: Table 1 in Supplementary PDF has some formatting problems.

Response

We thank the reviewer for noticing this, we have now made sure all submission material is well formatted.

Reviewer #2

This is a very clearly written paper describing a fairly large number of important analyses related to adherence to five commonly prescribed medication classes. The genetic results are especially important even though they suggest only modest contributions to adherence/persistence. The conclusions are well supported by the results. I have only minor comments overall:

Comment 1

This statement in the introduction is not entirely accurate: "Nonetheless, the impact of clinically relevant pharmacogenetic variants on drug adherence has not yet been studied." Perhaps the "clinically relevant" qualifier makes it less so but a quick Pubmed search showed several genetic studies of drug adherence. A few examples:

- Heritable Risk and Protective Genetic Components of Glaucoma Medication Non-Adherence
- Role of genetics in the prediction of statin-associated muscle symptoms and optimization of statin use and adherence | Cardiovascular Research | Oxford Academic
- Determinants of Sustained Viral Suppression in HIV-Infected Patients with Self-Reported Poor Adherence to Antiretroviral Therapy | PLOS ONE

Response

We thank the reviewer for the comment, and we agree that this statement is partially inaccurate as our study is not the first to investigate the impact of pharmacogenetic variants on drug persistence and adherence. We have now rephrased the statement at lines 71-72 to more accurately reflect the scope of our paper:

Nonetheless, the impact of clinically relevant pharmacogenetic variants on drug adherence has not yet been studied **in large biobank-based studies and considering multiple medications.**

Comment 2

This statement in the introduction has significant implications for the interpretation of this work: “We considered five commonly prescribed medications that are typically used for long-term, continuous, reg

ular therapy, and do not have directly observable effects on symptoms” and warrants further discussion, especially as it relates to side effects, which may have an equally if not greater effect on adherence, especially in terms of genetic effects. Doesn't uncontrolled, elevated blood pressure in particular have a noticeable effect to the patient? Wouldn't the same be true of a patient who develops an atrial fibrillation or embolism due to Warfarin non-adherence? Is the statement even relevant to the goals of the study? If not, it could be removed.

Response

We thank the reviewer for highlighting this statement and its implications in the interpretation of the manuscript. We agree with the reviewer that for most medications, effects of sub-optimal adherence or adverse drug reactions would be noticeable to the patient. Furthermore, we agree that our focus is on prolonged regular therapy regimens and that the last part of the statement is not particularly relevant to the goals of this work. We thus removed it at lines 87-88:

We considered five commonly prescribed medications that are typically used for long-term, continuous, regular therapy, ~~and do not have directly observable effects on symptoms~~

Comment 3

The SNPs/genes associated with the three nearly GWS hits that did not appear in Open Target Genetics still warrant discussion in both the Results and Discussion.

Response

We thank the reviewer for the valuable suggestion. We have now added a section in the Supplementary Note providing further characterization of each of the 4 GWS hits reported in Supplementary Table 10:

Characterization of variants genome-wide significantly associated with adherence and persistence

Supplementary Table 10 reports the 4 variants associated with either adherence or persistence at $P < 5 \times 10^{-8}$. We further characterized of these variants based on evidences reported in Open Target Genetics ⁶.

- *rs1339882991*, positively associated with both adherence and persistence to BP medications, is an intronic variant located in proximity of the *WNT2B* gene, showing a V2G assignment to the same gene based on evidence from brain tissue eQTLs. This variant was previously reported to be associated with increased risk of hypertension⁷ and higher blood pressure⁸.
- *rs111349244*, associated with lower odds of persistence to BP medications, is an intronic variant located near the *LINC02227* gene and was associated with a lower number of antihypertensive medication purchases and decreased risk of hypertension in FinnGen.
- *rs12149025*, associated with lower persistence to DOAC, is an upstream gene variant located in proximity of the *CBFA2T3* gene, with V2G assignment to *CDH15* based on PCHi-C⁹ evidences and evidences from muscle tissue eQTLs.
- *rs548379361*, associated with lower adherence to breast cancer medications, is an intronic variant near the *CFAP44* gene.

We also pointed to this in the Results section at line 225:

Further characterization of this and the other genome-wide significant variants is provided in the **Supplementary Note**.

Finally, we added the following sentence to the Discussion, at line 341:

We did not identify robust individual genetic signals but the identification of 4 genome-wide significant variants, 2 of which were not previously reported to be associated with traits underlying the need for medication, warrants for larger genetic studies of adherence and persistence.

Comment 4

Additional bioinformatics analysis of the top associations would improve the paper. Do the SNPs have regulatory potential, are they eQTLs/mQTLs, transcription factor binding sites, etc.

Response

We again thank the reviewer for the valuable suggestion, we added further characterization of the top associated variants in the Supplementary Note (see response to Comment 3).

Reviewer #3

The authors have undertaken a large-scale study using FinnGen and the Estonian biobank to evaluate factors (social, clinical and genetic) associated with poor adherence and persistence to medicines. They found that need for social assistance and immigration status showed a

negative effect on persistence and adherence – this has been identified before, and therefore not surprising. Investigation of genomic factors did not identify any meaningful associations, which is also not surprising.

Comments 1,2

Although the authors have done a lot of work on this paper, and the paper is well written, the only positive associations identified are not novel. Furthermore, the genomic investigations are unsurprisingly negative and probably reflect the tenuous underlying hypotheses.

As the authors acknowledge, just because somebody picks up their medicines from a pharmacy does not necessarily mean they are taking them. All studies have shown that adherence deteriorates the more medicines the person is supposed to be taking – this gets worse when you have to take the medicines more than once per day. Hence a major omission from the study is the lack of any assessment of the number of medicines, and how often per day, being taken by each individual which should be an important indicator of adherence.

Response

We thank the reviewer for the comments. We agree that drug purchase data only represents a proxy for actual drug-taking, although it is a far more accurate proxy than, for example, prescription data. We tried to minimize any discrepancies between purchasing medications and actually taking them by including in the adherence analysis only individuals purchasing the medication for at least 12 months and by excluding all individuals with excessive overbuying or stock-piling behaviors, for which it would be also difficult to estimate to which extent they are actually taking the medication. We acknowledge this a limitation but also notice that across all medications the average treatment length varies between 2.9 and 11 years, making it a reasonable assumption that individuals included in the study and purchasing medications for such a long period of time will also be taking it regularly.

Comment 3

I am also confused by the medications studied:

- Antihypertensives are classified separately from diuretics, calcium channel blockers, RA acting drugs (page 20, lines 442-443). What were the other antihypertensives?
- Why wasn't acetylsalicylic acid considered on its own? Seems to have been considered together with dipyridamole.
- Caplacizumab is not an anticoagulant.

Response

We thank the reviewer for raising these points and we provide better clarification for each of them. The medication classification we report follows the ATC classification system. For blood pressure medications, we included all drugs with ATC starting with C02* (antihypertensives), C03* (diuretics), C08* (calcium channel blockers) and C09* (RA acting agents). Specifically, the C02* (antihypertensives) class includes the following subclasses:

- C02A Antiadrenergic agents, centrally acting
- C02B Antiadrenergic agents, ganglion-blocking
- C02C Antiadrenergic agents, peripherally acting

- C02D Arteriolar smooth muscle, agents acting on
- C02K Other antihypertensives
- C02L Antihypertensives and diuretics in combination
- C02N Combinations of antihypertensives in ATC gr. C02

(medications included in each subclass can be found at https://www.whocc.no/atc_ddd_index/?showdescription=yes&code=C02).

Acetylsalicylic acid was not considered on its own because only reimbursable prescription medications are included in the drug purchase data, making it possible only to retrieve purchase of the ASA+dipyridamole combination.

Finally, we want to thank the reviewer for noticing we erroneously included caplacizumab as an anticoagulant. The mistake was due to looking up the wrong ATC code (B01AX07) instead of B01AE07, for dabigatran etexilate. We apologize for this mistake and we have now reported the right name in the **Methods** section.

We edited the **Methods** section (line 472) as follows, including the ATC codes next to each drug class name, and reporting the right name for ATC B01AE07:

The classes of medication we included are as follows: statins (C10AA* - all types of statins without differentiation), blood pressure medications (C02* - antihypertensives, C03* - diuretics, C08* - calcium channels blockers, C09* - agents acting on the renin-angiotensin system), antiplatelets (B01AC04 - clopidogrel and B01AC30 - acetylsalicylic acid in combination with dipyridamole), breast cancer medications (L02BA01 - tamoxifen, L02BG03 - anastrozole, L02BG04 - letrozole, L02BG06 - exemestane), direct oral anticoagulants (DOAC) (B01AF01 - rivaroxaban, B01AF02 - apixaban, B01AF03 - edoxaban, B01AE07 - dabigatran etexilate caplacizumab)

Comment 4

Also tolerability varies within each of these classes, and therefore classifying drugs as a class is likely to miss the effects of individual drugs on tolerability. Another major issue, highlighted by the authors, is inaccurate assessment of doses.

Response

We thank the reviewer for the comment and agree that looking at classes of medications might miss some individual drug effects related to tolerability and metabolism. For this reason, for the analyses where these aspects were extremely relevant to take into consideration, for example for the pharmacogene analysis, we looked at individual drugs rather than the class. For the other analyses, e.g. GWAS, we prioritized power for discovery by considering the class and having a bigger sample size. As for the assessment of daily doses, we acknowledge this as a limitation resulting from not having prescription indications available for each purchase. However, as prescription data is partially available as text starting in 2015, we checked this for

statins for the year 2019 and found that 96% of the prescriptions were indicating the expected daily dose of 1 statin tablet/day.

We would like to thank the reviewers for their thoughtful comments and suggestions. We have thoroughly addressed the issues raised by the reviewers. Our responses to the reviewers' comments are colored in blue in the following, while the new texts added to the manuscript are colored in red.

Reviewer #1 (Remarks to the Author):

The authors have addressed all my comments. I have no further questions.

We thank the reviewer for confirming that all comments have been addressed and for the thoughtful feedback provided.

Reviewer #2 (Remarks to the Author):

The authors fully addressed my minimal concerns with the paper and made appropriate changes to the text.

We thank the reviewer for confirming that all comments have been addressed and for the thoughtful feedback provided.

Reviewer #3 (Remarks to the Author):

The authors have made most of the changes following the last review. However, an important aspect NOT answered is the effect of the number of medicines taken per day on adherence (reviewer #3, comments 1,2). The applicants may want to look at these papers, and ideally include an assessment of medication burden on non-adherence in their paper.

Zelko E, Klemenc-Ketis Z, Tusek-Bunc K. Medication adherence in elderly with polypharmacy living at home: a systematic review of existing studies. *Mater Sociomed* 2016;28:129–32.

Foley L, Larkin J, Lombard-Vance R, et al. Prevalence and predictors of medication non-adherence among people living with multimorbidity: a systematic review and meta-analysis. *BMJ Open* 2021;11:e044987.

Response:

We thank the reviewer for the comment, and we apologize for not addressing this point in the previous revision. In our primary analyses, we have not included an assessment of the effect of polypharmacy on adherence and persistence as we focused on factors at treatment baseline, or factors fixed over time (genetics). We have now performed an additional analysis evaluating the

impact of polypharmacy on adherence and persistence in the nation-wide Finnish data. For each medication, we determined if any of the other four treatments was concurrent. We fitted a linear model for persistence and adherence with a categorical variable with three levels (0 for no concurrent treatment, 1 for one concurrent treatment, and 2 for more than one concurrent treatment), adjusting for the baseline factors included in our main analysis. Overall, we found a consistent positive effect of polytherapy on adherence and persistence.

We added the following sentence in the main text at line 171:

Extending beyond these baseline factors, we further assessed the effect of one or more concurrent treatments (i.e. polytherapy). We observed a consistent positive association between having at least one or more treatment and higher adherence and increased odds of persistence (**Supplementary Note**).

We also further detailed this analysis in the **Supplementary Note** at line 69, by adding the following paragraph:

Effect of polytherapy on adherence and persistence

We assessed the effect of polytherapy on drug adherence and persistence with respect to the five medications defined in the primary analysis. For each medication, we determined if any of the other four treatments were concurrent in the following manner: for adherence, we considered a medication regimen concurrent if the time between the first and last purchase recorded was overlapping at any time with the timespan used for adherence calculation; for persistence if the time between first and last purchase recorded of the potential concurrent treatment contained the purchase date used for persistence calculation. We fitted a linear model for persistence and adherence with a categorical variable with three levels (0 for no concurrent treatment, 1 for one concurrent treatment, and 2 for more than one concurrent treatment), adjusting for the baseline covariates used in the primary analysis (**Health and socio-demographic risk factors for persistence and adherence**). The percentages of change in adherence for polytherapy are reported and OR for persistence are reported in **Supplementary Tables 21-22**.

We observed that the presence of at least once concurrent treatment was consistently associated with both increased adherence and higher odds of persistence. The percentage increase in adherence with one concurrent treatment ranged from 0.6% (blood pressure medications) to 3.6% (antiplatelets). The ORs of being persistent between one concurrent treatment and no concurrent treatments go from 1.04 (not statistically significant for breast cancer medications) to 2.79 (anticoagulants). Moreover, we find consistently larger effect sizes for two or more concurrent treatments compared with only one concurrent treatment, except for blood pressure medications.

We would like to thank the reviewers for their thoughtful comments and suggestions. We have thoroughly addressed the issues raised by the reviewers. Our responses to the reviewers' comments are colored in blue in the following, while the new texts added to the manuscript are colored in red.

Reviewer #3:

“The authors have undertaken a further analysis on polypharmacy and adherence. They have included a couple of lines in the main paper, and included supplementary details. The authors find a positive "association between having at least one or more treatment and higher adherence and increased odds of persistence". However, there is no discussion of this finding especially since it is discordant with other literature on adherence which shows that adherence worsens as the person has to take an increasing number of medications. Why do the authors think their results provide an effect which is discordant with other literature?”

Response:

We thank the reviewer for the comment and we agree that further elaboration about this point is needed. We added the following sentence to the Discussion section, elaborating on the possible reasons for the positive effects of polypharmacy on adherence and persistence we observed in our study, and mentioning previous literature highlighting inconsistent effect of polypharmacy:

“Similarly, the concurrent use of multiple medications was also associated with increased adherence and odds of persistence, likely capturing an underlining perceived increased risk, as it is likely the case also for secondary prevention. Although previous literature showed inconsistent or negative effects of multiple concurrent treatments on adherence, we attribute the positive effect observed in our data to the fact that we are considering a restricted set of medications, with relatively high adherence rates.”